# Exploring the Link Between Obligate Anaerobe-Related Dysbiosis and Prostate Cancer Development: A Pilot Study

**DOI:** 10.3390/cancers17010070

**Published:** 2024-12-29

**Authors:** Efthymios Ladoukakis, Tim Oliver, Mark Wilks, Emily F. Lane, Frank Chinegwundoh, Greg Shaw, Belinda Nedjai

**Affiliations:** 1Wolfson Institute of Population Health, Faculty of Medicine and Dentistry, Queen Mary University of London, London EC1M 6BQ, UK; m.ladoudakis@qmul.ac.uk (E.L.); e.lane@qmul.ac.uk (E.F.L.); 2Barts Cancer Institute, Faculty of Medicine and Dentistry, Queen Mary University of London, London EC1M 6AU, UK; r.t.oliver@qmul.ac.uk; 3Blizard Institute, Faculty of Medicine and Dentistry, Queen Mary University of London, London E1 2AT, UK; m.wilks@qmul.ac.uk (M.W.); frank.chinegwundoh@nhs.net (F.C.); 4Division of Surgery and Interventional Science, University College London, London WC1E 6BT, UK

**Keywords:** prostate cancer, anaerobic bacteria, immune deficiency, circumcision

## Abstract

Prostate cancer (PCa) is associated with the inflammation caused by anaerobic bacteria such as *Cutibacterium acnes*. In a previous exploratory study using 39 participants and a conventional bacterial culture approach (culturomics), we discovered a link between obligate anaerobic bacteria and increased prostate-specific antigen (PSA), a biomarker for prostate cancer risk. Our current study further validates this discovery by examining the microbial content in urine samples from 89 PCa patients using the more sensitive approach of 16s rDNA sequencing. The results show a clear association between specific obligate anaerobic bacteria such as *Peptostreptococcus* and increased PSA levels. Our findings indicate a potential contribution of obligate anaerobic bacteria in chronic inflammation that can lead to precancerous lesions, but larger cohorts are needed to further evaluate their role in prostate cancer development and progression.

## 1. Introduction

Prostate cancer (PCa) is the second most frequently diagnosed cancer in men and the fifth leading cause of death worldwide [1]. However, when stratifying by country, prostate cancer becomes the most frequently diagnosed cancer in 118 countries and the leading cause of death in 52, mostly in low- and middle-income countries (LMICs) [2]. This unequal distribution of the disease prevalence suggests global disparities in early detection and successful treatment. In addition to the socioeconomic reasons behind prostate cancer’s prevalence, its unique pathophysiology is also a deciding factor. Prostate cancer may be asymptomatic at the early stage and often has an indolent course, which leads to minimal or even no treatment for many years. When symptoms occur, they most often include difficulty with urination, frequent urination, or nocturia, which are also common in benign prostatic hypertrophy (BPH) [3].

One of the most established and recommended screening tests for the early detection of prostate cancer is the prostate-specific antigen (PSA) test [4]. PSA is an enzyme secreted by the prostate epithelium, whose function is to break up large proteins found in the semen in order to decrease seminal viscosity and facilitate sperm motility. Elevated levels of PSA have been associated with prostatic adenocarcinoma and have been used as a risk factor in the clinic for both disease onset and severity, though the PSA test’s low specificity can lead to overdiagnosis and overtreatment [5,6]. This has led to the pursuit of novel and more accurate biomarkers for prostate cancer early diagnosis. Among these, the urinary microbiome has been shown to be linked to the disease, although the underlying mechanisms are yet to be fully elucidated [7].

Increasing evidence [8,9,10,11,12] has suggested that prostate cancer may follow a microbiome-based etiological mechanism similar to the one linking infection with the microaerophilic (MA) *Helicobacter pylori* to gastric cancer [13,14]. Over the past few years, it has become clearer that there are bacterial infections beginning in puberty, other than *H. pylori*, that are involved in the pathogenesis of cancer, and this is opening up new ideas on the pathogenesis of PCa. It is now 69 years since the first pathologist reported that areas of stromal atrophy represented a possible precursor associated with latent prostatic cancer [15]. In 1996, Feneley et al., using Ki67 in a prostate cancer study, demonstrated that neighboring normal prostate cells in these atrophic foci had almost as high a mitotic index as in the nearby cancer. In 1999, de Marzo et al. from the Johns Hopkins group coined the expression “proliferative inflammatory atrophy” because of the evidence of inflammatory changes associated with the atrophic foci and postulated that the high proliferative rate ultimately resulted in generation of in situ carcinoma [16,17].

One particular example is the puberty-associated presence of the microaerophilic bacterium *Cutibacterium acnes*,which has been linked to PCa in several studies over the last 17 years [18,19,20,21,22,23]. This evidence has led to a renewal of interest in the association between the dysbiosis of bacteria and PCa as well as a consequent increase in the study of the urinary microbiome and its link to sexually transmitted disease [24]. While the dysbiosis of anaerobe/microaerophilic bacteria has been well established as a driver for PC, the effect of obligate anaerobes (OAs) is yet to be thoroughly investigated for this disease. However, several studies have shown associations between specific anaerobic bacteria and various others cancers, including colorectal and cervical cancer [25,26]. The rationale behind this study and the data presented here are further supported by a study from 1988 by Cooper, P. et al. linking anaerobes to PCa, suggesting other anaerobic organisms in addition to *C. acnes* might have similar carcinogenic ability [27]. This led to our previous study using a bacterial culturomics approach with conventional agar culture followed by matrix-assisted laser desorption ionization time-of-flight (MALDI-TOF) identification of growth while studying post-rectal examination urine from PCa patients on active surveillance and in patients with suspicious PCa urinary symptoms [28,29,30]. A recent review of possible mechanisms has supported our report and suggests that these findings could be linked to perturbation of the gut microbiome [31,32,33].

Our current analysis updates the results from our previous culturomics study using a cohort of 39 patient samples as well as report results from the more sensitive technique of 16S rDNA sequencing in 89 patient samples from the PROVENT trial (ClinicalTrials.gov ID: NCT03103152) [34]. In this pilot study, we sought to explore the link between OA-related dysbiosis and PCa development with the goal to contribute to novel prevention and treatment strategies similar to the ones targeting *H. pylori* for gastric cancer treatment [35,36].

## 2. Materials and Methods

### 2.1. Culturomics Cohort

This cohort was composed of two populations. The first population initially reported in 2017 [29] and studied between 1 January 2015 and 14 September 2015 included adult male patients aged >18 years old with a histological diagnosis of prostate cancer with a Gleason grade ≥ 3 + 3 and on active surveillance while awaiting a decision on the need for radical therapy (*n* = 18). The second population had urinary symptoms in need of investigation (*n* = 21) and results were only reported in a meeting abstract [30]. For each sample, 1.5 mL of urine was centrifuged at 16,000× *g* for 2 min. The supernatant was then discarded, and 10 μL of the deposit was plated on Columbia Horse Blood agar (Thermofisher, Basingstoke, UK) and Fastidious anaerobic agar (Thermofisher, Basingstoke, UK). Plates were incubated for 72 h at 35 ˚C in air  +  5% CO_2_ or anaerobically in an atmosphere of 90% N_2_, 5% H_2_, and 5% CO_2_. Resultant colonies were identified by MALDI-TOF using a Bruker Maldi Biotyper mass spectrometer [37].

### 2.2. 16S Sequencing Cohort

This cohort comprised 89 male patients aged >16 years old with a histological diagnosis of prostate cancer with Gleason grade 6 or 7 and no previous treatment for prostate cancer. These patients were recruited to PROVENT, a phase 2 placebo-controlled trial of two doses of aspirin (100 and 300 mg) and one dose of Vitamin D (4000 IU in nasal drops daily) [34]. Patients recruited from prostate clinics at Barts and Newham General Hospitals had signed informed consent for storage of clinical information and research on their urine sample after rectal examination under ethics approval (REC reference 17/LO/0109). Between 1 February 2020 and 12 February 2020, all case histories were reviewed for follow up by one of the authors (TO). All PROVENT trial samples (*n* = 89) were collected at baseline during the trial under protocol (ISRCTN registry ID: ISRCTN91422391), approved by the trial sponsor (Queen Mary University of London), the Integrated Research Application System (IRAS projects ID 145427 and ID: 21506), and the National Research and Ethics Service (REC reference 14/LO/2033) and implemented in accordance with Good Clinical Practice guidelines and the Declaration of Helsinki.

### 2.3. Bioinformatic Analysis (16S Sequencing Cohort)

#### 2.3.1. Amplicon Sequence Variants

Raw 16S sequencing data were produced in two batches of 50 and 39 samples and were pre-processed separately by Clinical Microbiomics. The preprocessing included a customized pipeline in R based on the *dada2* package, which provided amplicon sequence variant (ASV) abundance tables as well as ASV read count tables. During preprocessing, primer sequences were removed from raw reads using *cutadapt* [38]. Reads without a primer matches or with ambiguous bases (e.g., Ns), as well as reads shorter or longer than expected from the number of sequencing cycles and the lengths of the primers, were filtered out. In an additional filtering and trimming step (*filterAndTrim* command), reads were trimmed at the 3′ prime end based on sample-specific quality scores. Trimmed reads expected to contain multiple sequencing errors (>2.2 for forward reads and >2.6 for reverse reads) based on their nucleotide quality scores were removed. The remaining reads were de-replicated into unique sequences (*derepFastq* command), and forward and reverse reads were then denoised separately for each sample (*dada* command). During denoising, less abundant sequences closely related to more abundant sequences could be reassigned, assuming they were sequencing errors of the more abundant sequence. Denoised forward and reverse reads were merged (*mergePairs* command), and read pairs were only kept if they had sufficient overlap without mismatches, had a merged sequence within the 400 to 432 bp V3V4 amplicon size range, and satisfied one or more of these criteria: a relative abundance of 0.02 or a count of 700 in any sample, or occurrence in at least 4 samples. Finally, suspected bimeras (two-parent chimeras) were removed from the generated abundance table through internal abundance and sequence comparisons (*removeBimeraDenovo* command).

#### 2.3.2. Taxonomic Analysis

The taxonomic assignment of the detected ASVs for both sequencing batches was also performed by Clinical Microbiomics and included 2 steps per batch. First, the ASV sequences were compared to full-length 16S sequences in an internal reference database (CM_16S_27Fto1492R_v1.0.0) using a naïve Bayesian classifier. The reference database was generated using in silico extraction from the GTDB database (release: 06-RS202) [39], the rrnDB (version 5.7) [40], and the UHGG database (v2.0) [41] and subsequent curation. In the second annotation step, the taxonomic assignments were improved using precise sequence identity percentages between the found ASVs and amplicons in an internal V3V4 amplicon database (CM_16S_341Fto785R_v1.0.0.rds). The final abundance and read count tables that were exported included a full taxonomic characterization from domain to species for each ASV (Appendix A).

#### 2.3.3. Data Merging and Downstream Analysis

In order to further analyze the pre-processed data, they were transferred to QMUL’s High Performance Cluster Apocrita [42] and imported in an R environment using the *Phyloseq* package [43]. The read count table along with the taxonomic information were used to create two separate *Phyloseq* data objects (one per batch). Age, PSA levels, and sequencing batch were added to the data objects as additional metadata, with the last used for downstream batch correction. Before merging the two objects (*merge_phyloseq* command), ASV sequences were used to identify common ASVs between the two batches. In the merged object, taxa were agglomerated (*tax_glom* command) in the ‘Genus’ level. Normalization was achieved by transforming raw read counts to relative abundances. Non-metric MultiDimensional Scaling (NMDS) based on Bray–Curtis distance indices [44] was used via the *Phyloseq* package after taxa agglomeration and normalization to confirm no batch effects being present. *Phyloseq* was also used to estimate alpha and beta diversity for all the samples. Alpha diversity was estimated using the Shannon index [45]. For the display of beta diversity, only the top 20 most abundant genera were kept after removing the unclassified genera. A heatmap of the top 20 most abundant genera and their relation to PSA was created via the *Pheatmap* package in R.

### 2.4. Statistical Assessment

#### 2.4.1. Culturomics Cohort Analysis

In order to determine whether PSA level was associated with the presence of either MA or OA bacteria, a Wilcoxon-type test for trend, the Cuzick test, was carried out with “neither” as the reference group using the *rawr* package in R. Welch’s two-sample *t*-tests were also used to identify differences in PSA between OA-positive and -negative samples.

#### 2.4.2. 16S Sequencing Cohort Analysis

PROVENT samples were stratified into two groups based on the median PSA of the cohort, resulting in the High-risk group (PSA > median PSA) and the Low-risk group (PSA ≤ median PSA). For the differential abundances of ASVs, Linear Discriminant Analysis Effect Size (LefSe) was performed using the microbioMarker package with a stringent LDA cutoff threshold of 3 to identify genera that differ significantly between the Low-risk and High-risk groups [46]. NMDS plots were created via the *Phyloseq* package to identify global differences between the two risk groups. Welch’s two-sample *t*-tests were also used to identify differences in alpha diversity between the two risk groups.

Due to the data from the earlier culturomics study suggesting that OAs were associated with a higher level of PSA, analysis of the PROVENT data was initially restricted to OA, and the effects were also analyzed by group rather than individual species. As such, a second sample stratification was performed based on whether the pooled relative abundance of OA genera was above a certain threshold (OA-positive status). Three different pooled abundance thresholds were examined—10%, 15%, and 20%—to split the samples into OA-positive and OA-negative groups. OA status was defined using total abundance of *Finegoldia*, *Fusobacterium*, *Prevotella*, *Peptoniphilus_A*, *Peptostreptococcus*, and *Veillonella_A*,as these were the OA genera identified both in the 16S sequencing and in the culturomic cohort. An additional stratification was performed with the addition of three other obligate anaerobes, related but different genera that were detected in the 16S sequencing cohort but not in the culturomic study (*Peptoniphilius_B*, *Peptoniphilius_C* and *Veillonella*). Welch’s two-sample *t*-tests were used to identify differences in PSA between OA-positive and -negative samples. Moreover, to identify statistically significant associations between risk group and OA status, Fisher’s exact test was used.

A final stratification was performed based on whether the relative abundance of a single specific OA genus, from the six of interest, was above a certain threshold. Three different abundance thresholds were examined—0%, 5%, and 10%—to split the samples into OA-positive and OA-negative groups. Similar to the pooled abundance stratification method, Welch’s two-sample *t*-tests were used to identify differences in PSA between OA- positive and -negative samples.

Besides focusing on the OA of interest from the culturomics cohort, a data-driven approach was employed to uncover potential taxa that were associated with PSA. Pearson correlation coefficients were measured between each of the genera abundances and PSA levels. Multiple linear regression analysis was performed, including the genera that produced a statistically significant correlation between relative abundance and PSA and the original six genera from the culturomics cohort.

All statistical comparisons were performed in the same R environment, which was used to load and manipulate the sequencing data.

## 3. Results

As mentioned above, in order to evaluate the role of OA in the progression of prostate cancer, we performed two distinct analyses using data from two separate cohorts; the first one was the culturomics cohort using MALDI-TOF methodology, and the second was the PROVENT cohort using the more sensitive 16S rDNA sequencing technique.

### 3.1. Culturomics Cohort Results

Urine samples were collected from 39 patients diagnosed with biopsy-proven or suspected prostate cancer from the Barts and Newham hospitals, and their PSA levels were recorded from blood samples that were taken concurrently. The OA and MA bacteria detected using the MALDI-TOF methodology are summarized in Table 1.

Of the 39 patient urine samples, twelve (30.7%) contained multiple OA species, up to four species per positive sample (mean = 1.7 per positive sample). Six urine samples (17.9%) contained MA bacteria species, up to two per positive sample (mean = 1.2 per positive sample). In the OA-positive patients, PSA levels had a mean of 9.25 and median of 6.75 ng/L; in the MA-positive patients, PSA levels had a mean of 6.51 and median 6.34 ng/L, while in 22 patients without any anaerobes the mean was 4.75 and the median 3.40 (see Table 1 and Appendix A).

There was a generally lower level of PSA in the urinary symptoms group from Newham as only three had cancer. The remainder had benign prostatic hypertrophy but with symptoms or raised PSA, justifying biopsy, while all 18 in the Barts series had cancer detected and were on active surveillance.

When the samples were ranked based on their bacteria composition (OA-positive, MA-positive, OA/MA-negative), a statistically significant trend was observed (Cuzick test *p* = 0.034, Table 2) for PSA levels. The lowest median of PSA levels was observed in the group of urine samples without OA/MA, an elevated median in samples with MA species, and slightly more elevated PSA median in samples with OA species.

Patients from the culturomics cohort were followed up for a median period of 43 months. Overall, ten out of the seventeen patients (58.8%) with MA and/or OA compared to five out of twenty-two (22.7%) without MA/OA needed a urological intervention during the period of observation (Appendix A, *p* = 0.045, Fisher’s exact test).

### 3.2. 16S Sequencing Cohort Results

Since the data from Table 1 and Table 2 demonstrated that OA produced a statistically significant increase of PSA, we decided to evaluate further this initial finding using 16S rDNA sequencing on the PROVENT cohort. Our initial focus was on the same group of OA genera (*Finegoldia*, *Fusobacterium*, *Prevotella*, *Peptoniphilus_A*, *Peptostreptococcus*, and *Veillonella_A*). Three of the six genera (*Peptoniphilius_A*, *Prevotella*,and *Veillonella_A*) were in the top 20 genera, ranked based on their relative abundance. Alpha and beta diversity per sample for these top 20 genera are displayed in Appendix A. Hierarchical clustering of the samples based on their relative abundance of the top 20 genera did not reveal any clusters associated with any of these two PSA risk groups (Appendix A).

Stratification of the 89 PROVENT samples based on the median PSA of the cohort resulted in two groups, the High-risk group (mean/median PSA = 10.14/9.80 ng/mL) and the Low-risk group (mean/median PSA = 4.84/5.10 ng/mL). To evaluate the microbiome on these samples, we first calculated the alpha diversity per sample and then compared the alpha diversity (Shannon Index—H’) between the two risk groups (Figure 1). This showed a slightly smaller median alpha diversity in the High-risk PSA group, but no statistical significance was observed (ΔH’ = 0.144, *p* = 0.316).

NMDS analysis is an ordination technique similar to principal component analysis (PCA) and was used to visualize the similarities of the samples in each of the two risk groups based on their microbial abundance. No clustering of samples could be observed as associated with the high/low-risk groups (Figure 2), which means that any differences between the two groups are not driven by a global difference in microbial composition but rather are limited to specific bacteria.

LEfSe analysis was implemented to uncover the specific bacteria that were statistically enriched in the high/low-risk groups. This statistical test uses the Kruskal–Wallis rank-sum test to detect features (i.e., microbial genera) with statistically significant differences between groups (i.e., high/low-risk groups) and calculates the effect size of each feature using linear discriminant analysis (LDA). It identified 13 enriched genera in the Low-risk group and 8 genera in the High-risk group (Figure 3), including *Peptostreptococcus* and Fusobacteria from the six bacteria identified in the culturomics cohort.

As mentioned before, the six OA genera from the culturomics cohort were used for our second stratification of the PROVENT samples into OA-positive and OA-negative groups (Table 3). For each sample, this grouping depended on whether the pooled relative abundance of these six OA genera was above or equal a specific threshold, as opposed to the culturomics cohort, where the classification was based on whether one or more OA bacteria were detected. When using the ≥10% pooled relative abundance threshold, 21/89 (23.6%) samples from the PROVENT cohort were considered as OA-positive compared to 12/39 (30.8%) in the culturomics analysis. For that threshold, mean PSA was higher in the OA-positive group, but with no statistical significance (*p* = 0.133). The difference in mean PSA was higher between groups when the threshold was increased to ≥15% with a lower *p*-value (*p* = 0.082). When the threshold was increased even further, to ≥20%, the difference in mean PSA across groups was even higher and also statistically significant (*p* = 0.006). When this analysis was repeated using the addition of three other related but different genera, not detected in the initial culturomics analysis (*Peptoniphilius_B*, *Peptoniphilius_C* and *Veillonella*), the ≥20% threshold was still the one that displayed statistically significant differences of mean PSA between the two groups (Appendix A).

Table 4 reports a cross-tabulation analysis of high/low PSA groups and the presence or absence of OA relative abundance. Fisher’s exact test detected statistically significant and increasing association between OA-positive samples and high/low-risk groups for all pooled relative abundance thresholds of 10%, 15%, and 20%. From these tables, it is evident that when a sample is OA-positive, it is more likely to fall into the High-risk PSA group. This likelihood increases as the pooled relative abundance threshold increases from 10% to 20%.

Table 5 investigates the effect of the six genera individually at ≥10%, ≥5%, and >0% relative abundance. Only at >0% relative abundance is there a significantly higher PSA linked to the expression of *Peptostreptococcus* (*n* = 9, *p* = 0.033) and Fusobacteria (*n* = 10, *p* = 0.009). At >0 relative abundance, there was a lower PSA in >0 relative abundance positive for *Prevotella* (*n* = 44). However, at ≥5% and ≥10% relative abundance, there is a higher, though non-significant PSA increase. We observed that all nine *Peptostreptococcus*-positive samples were positive for *Prevotella*,and five of them were positive for both *Prevotella* and *Fusobacterium*.

In order to further verify these observations in the context of the complete data from the PROVENT study, we performed Pearson correlation analysis between PSA and the relative abundances (>0 relative abundance prevalence) of all the genera. Moderate to strong Pearson correlation coefficients (−0.813 < r < 0.791) with *p*-values < 0.1 were observed between PSA and the relative abundances of seven genera (Figure 4).

The highest positive correlation was with *Peptostreptococcus* (r = 0.791, *p* = 0.011) (*n* = 9/89) and the lowest negative correlation was with aerobic *Methylobacterium* (r = −0.813, *p* = 0.002) (*n* = 11/89). Other positive correlations were identified with aerobic *Ochrobactrum_A* (r = 0.586, *p* = 0.027) (*n* = 14/89), and with OA *Prevotella* (r = 0.346, *p* = 0.021) (*n* = 44/89). A diagonal correlation matrix including the correlation coefficients between the relative abundances of different genera is displayed in Appendix A. As *Methylobacterium* and *Ochrobactrum_A* were identified as potentially useful for further study, they were added in a multiple linear regression model together with the six OAs previously observed to determine their individual effect on PSA levels. The results from this model (Table 6) showed that only *Peptostreptococcus* and *Ochrobactrum_A* were associated with a statistically significant positive effect on PSA (*p* = 0.007 & *p* = 0.009). The presence of *Prevotella* was associated with a non-significant positive effect (*p* = 0.236) whereas *Methylobacterium* had a non-significant negative effect (*p* = 0.079).

## 4. Discussion

The last 17 years has seen at least six publications [18,19,20,21,22,23] linking puberty-associated presence of an anaerobe/microaerophilic bacterium, *Cutibacterium acnes* and PCa. Three provide the most significant insight. Ugge et al. [18] reported a cohort study of 243,187 military recruits followed for 36 years. Medical records at the time of recruitment (average age of 21) indicating clinically significant acne had an HR of 1.43 (1.06–1.92) of subsequently developing PCa. There was stronger association with advanced prostate cancer (HR 2.37, 1.19–4.73). In the second insight, Mak et al. [21] reported that 30% of prostatectomies performed for cancer, cultured before fixation, had detectable *C. acnes*. The third insight came from two reports by Brae et al. [23] and Kakegawa et al. [22] using immunofluorescent anti-*C. acnes* monoclonal antibody on prostatectomy specimens. They reported 14.3% of the peripheral zone glands from 28 PCa patients positive for *C. acnes* compared to 4.3% *C. acnes* in 18 controls with bladder cancer (*p* < 0.001). They then followed up 80 raised PSA patients but negative prostate biopsy [22]. After an average follow up of 3 years, 44 developed PCa. These had 12.1% *C. acnes* positive glands in their initial biopsy compared to 4.8% in those who remained cancer negative on re-biopsy [22].

The data reported in this article lead to three observations of importance. The first is that the microaerophilic anaerobes had a lesser and non-significant effect on PSA compared to obligate anaerobes (Table 2 and Table 3). The data reported from the six cohort studies mentioned above suggests that microaerophilic bacteria could be present from puberty; this might be because they need to establish persistent infection and a minor degree of tissue damage to avoid rejection, preparing the environment for subsequent infections, which do more extensive damage. A recent study has suggested that there may be an additional mechanism that ensures the persistence of microaerophilic bacteria like *C. acnes* by invading epithelial cells, interfering with homologous repair and Fanconi anemia pathways through downregulating the expression of BRCA2 [47].

The second observation not previously reported was the wide range in the significance of relative abundance on the impact on PSA levels for different genera. The alpha diversity of the 89 samples from the PROVENT cohort, while showing considerable variation, was not significantly lower in individuals with elevated PSA levels. This suggested a potentially limited set of statistically associated bacteria, similar to findings in larger studies, and led us to focus the analysis on the six OAs reported in the culturomics cohort. From this focused analysis, we observed that some OAs at a low relative abundance (*Peptostreptococcus* and *Fusobacterium;* see Table 5) were significantly associated with a high PSA level, while others at a higher relative abundance were not (*Prevotella* and *Peptoniphilus_A*; Appendix A). Though neither of the latter pair were statistically significant, all nine of the low relative abundance *Peptostreptococcus* individuals were also positive for a high relative prevalence of *Prevotella*. Out of the 10 *Fusobacterium*-positive samples, four were also positive for *Prevotella*,while five were positive for both *Prevotella* and *Peptostreptococcus*. This suggests that there could be interaction between *Prevotella*, *Peptostreptococcus*,and *Fusobacterium*. The beta diversity of the top 20 by relative abundance (Appendix A) demonstrated a wide range of bacteria. However, they did include three OAs found to be linked with an increased PSA in the initial culturomics cohorts (*Prevotella*, *Peptoniphilus_A*,and *Veillonella_A* (Table 1)) and two of the remaining three were linked to increased PSA in the LEfSe analysis (*Preptostreptococcus* and *Fusobacterium*—Figure 3.

Our third observation was the presence of *Ochrobactrum_A*,which produced a statistically significant result in both the Pearson correlation analysis with PSA and the multiple linear regression analysis. This is a low relative abundance aerobic organism that belongs to the family *Brucellacae* and so far has only rarely been reported in microbiome cancer studies. These studies need extending to investigate the role in low relative abundant organisms of virulence factors such as the pathogen-derived genotoxin, colibactin, previously reported by Shrestha et al. [48], as they might be targetable by an appropriate vaccine. For the high relative abundance OA organisms, *Prevotella*, also linked to PCa in other reports [49,50] and systematic reviews [33] as well as to other cancers [51,52,53], their presence could be a reflection of the greater extent of direct bacterial organ cellular damage in low oxygen hypo-vascular tissue. An alternative mechanism is suggested in human and animal studies that have demonstrated that one *Prevotella* species, *Prevotella copri*, exhausts the levels of an intrinsic anti-cancer agent reagent called indole-3-pyruvic acid (IPyA) [52]. This lowering of IPyA levels results in breast cancer progression due to the phosphorylation of AMPK1 via increased transcription of UHRF1. The latter is associated with a DNA methylation inheritance mechanism and clearly needs further investigation, as it promotes cell differentiation and tumor suppression via epigenetic cell reprogramming when being inhibited [54].

In summary, the most important finding of our study is the confirmation of a link between obligate anaerobes and elevated PSA levels, which may indicate not only an increased risk of prostate cancer incidence but also greater cancer aggressiveness [55]. The strength of this study is that, in addition to validating the initial findings from the culturomics cohort, the rest of our results were derived using a strict data-driven approach. For instance, LEfSe analysis (Figure 3) was performed taking into account all the different species identified in the PROVENT samples, thus providing an unbiased representation of only the ones specifically associated with higher PSA levels. Similarly, Pearson correlation analysis (Figure 4) was performed between PSA and each one of the different genera detected in the samples, irrespective of its presence in the culturomics cohort. The reported results are the ones that displayed a relatively low *p*-value, providing potential candidates to add into our multiple regression analysis (Table 6).

A potential limitation of this study is the relatively small sample size of each cohort (Culturomics cohort: *n* = 39; PROVENT cohort: *n* = 89). In the case of the PROVENT cohort, the limited sample size restricted our multiple linear regression analysis with PSA to the main effects, when in fact the inclusion of interaction terms could potentially provide insight into the combinatory effect of specific bacteria, an effect closer to the biological reality.

Despite this limitation, our study establishes a direct link between obligate anaerobes and prostate cancer, paving the way for future research focused on specific bacteria using larger cohorts. Validating our findings in larger cohorts may identify additional key obligate anaerobe bacteria and uncover interactions between them, thus providing a mechanism with which microbiome dysbiosis contributes to prostate cancer. This may have considerable implications on prognostic and diagnostic strategies for the disease. For example, its long asymptomatic period may be addressed by the detection of potentially “high-risk” bacteria in the urinary microbiome. Such an approach may lead to novel prognostic tests and preventative strategies, offering new alternatives for the diagnosis and management of the disease.

## 5. Conclusions

The role of chronic inflammation and prostate cancer has been studied extensively [56], and the Johns Hopkins group has played a major role in demonstrating the complex interaction between bacteria including *C. acnes* [21,57,58], genetics [59] and immune-related cytokines [60]. More recently, they have demonstrated that bacterial foci in prostates produce pathogen-derived genotoxin colibactin. This toxin was able to induce DNA breaks in clinical material as well as cell culture experiments [48]. This work has contributed substantially to understanding the concept of the carcinogenic power of 20–50 years of chronic inflammation. Given that prostate cancer is not the only malignancy associated with long-term chronic inflammation and microbiome changes, targeting this inflammation could turn out to equal the data on stopping cigarette smoking, which starts in puberty; though this can produce death from lung cancer 20–40 years later [61], any prolonged stopping is linked to a reduced risk of death.

In following up a brief report from 2001 linking raised PSA to high titres of anti-chlamydial antibody in young gold miners attending a STD clinic in South Africa [9], Sutcliffe and colleagues from the Johns Hopkins group, in a series of reports starting in 2006 [62,63], found statistical significance between a history of three types of sexually transmitted infection (chlamydia, gonorrhoea, and non-chlamydia, non-gonorrhoea urethritis) and a raised PSA, which persisted when followed up, up to 10 years after initial screening [63]. Because the most significant effect on PSA came from the cumulative effect of all STDs including viruses, these data possibly suggest an environmental effect, such as a Vitamin D deficiency-induced chronic innate immune dysfunction or genetic immune hypo-function due to inherited immune response genes, explaining the infection susceptibility.

In the data reported in this paper, one third of the anaerobe-positive patients in the Barts and Newham Cohorts (Appendix A) did not have cancer. The anaerobe-positive non-cancer patients had a higher PSA than the remaining non-cancer samples from anaerobe-negative patients, and the former also had 11% more transurethral resections (TURs). This could be because chronic inflammation associated with the anaerobe infection could have preceded the prostate cancer, as suggested by the Johns Hopkins group. This pre-cancer occurrence of a pathogenic anaerobe has also been found in studies linking *H. pylori* with gastric cancer. Given the problems of increasing antibiotic resistance, focus on the future should be on immunology research aimed to unblock the “tolerance” mechanisms used by the anaerobic bacteria to escape rejection, possibly by use of short courses of checkpoint inhibitors or other forms of immunotherapy. Evidence that *H. pylori* induces PD-1 and PD-L1 expression in tumor cells [64] would support this approach. This could be facilitated by using simple clinical monitoring for inflammatory markers such as sedimentation rate, c-reactive protein, and red cell distribution width [65] as well as levels of immune cytokines [66] such as IL-6, IL-10, and gamma interferon. If this proved successful, this would considerably improve the selection of patients who might benefit from early radical local therapy and those safe to be offered active surveillance. Equally, in patients with locally advanced disease, the continued presence of anaerobes could accelerate clonal evolution of cancer by facilitating a more rapidly metastasizing disease. Such non-antibiotic therapy might also enhance the outcomes of surgery +/− pre-treatment short courses of radiation combined with checkpoint inhibitors to target bacterial and cancer radiation-induced antigens [67,68].

## Figures and Tables

**Figure 1 cancers-17-00070-f001:**
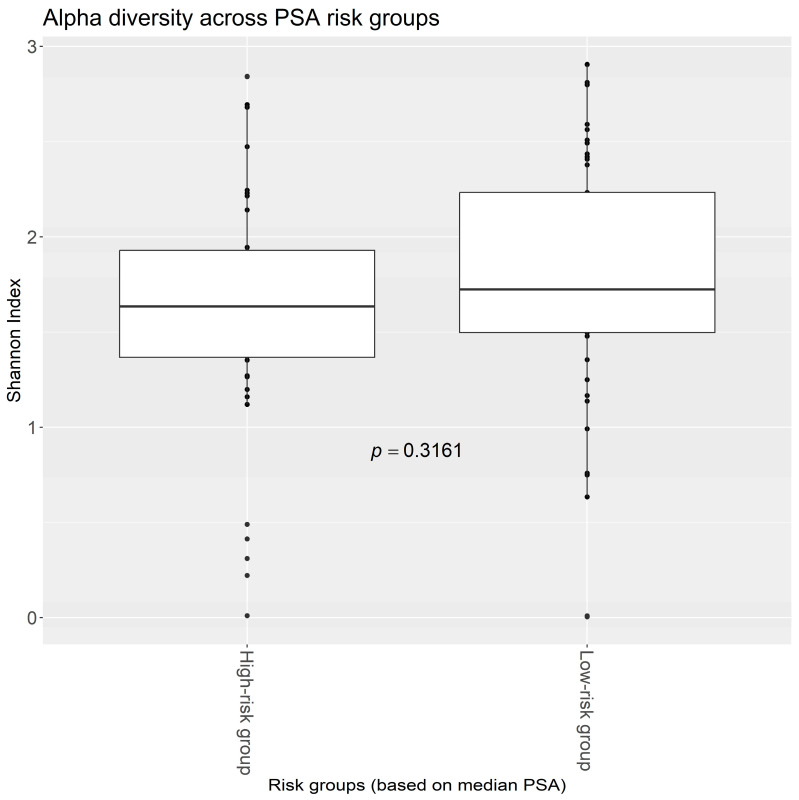
Alpha diversity between risk groups (Shannon index). The displayed *p*-value is from Welch’s two-sample *t*-test.

**Figure 2 cancers-17-00070-f002:**
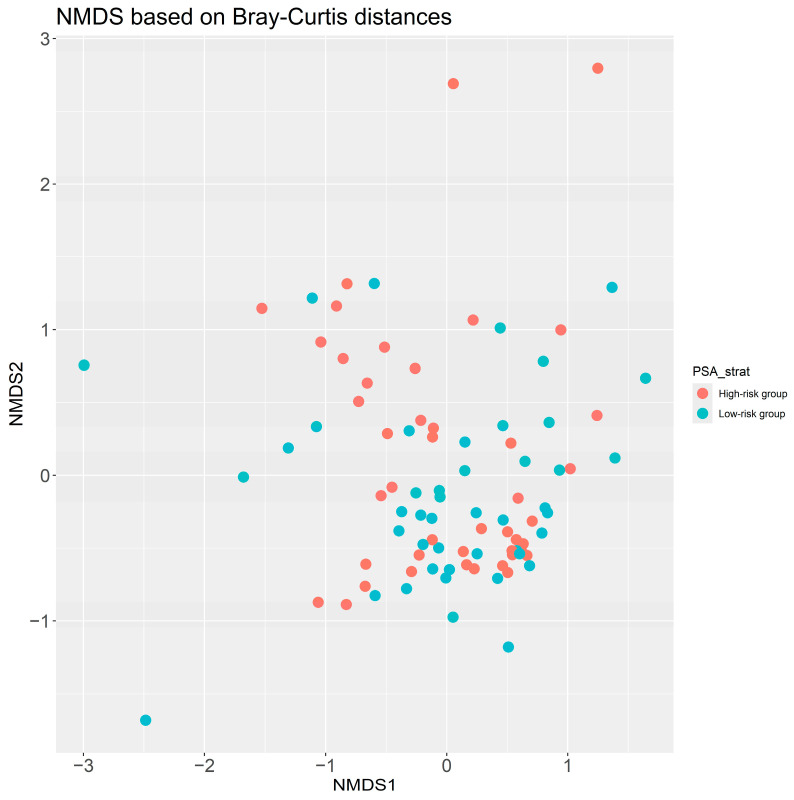
Non-metric Multidimensional Scaling (NMDS) analysis based on Bray–Curtis distances. Stratification of samples to high/low-risk PSA groups was done based on median PSA of the PROVENT samples: PSA > 7.1—High-risk PSA group, PSA ≤ 7.1—Low-risk PSA group.

**Figure 3 cancers-17-00070-f003:**
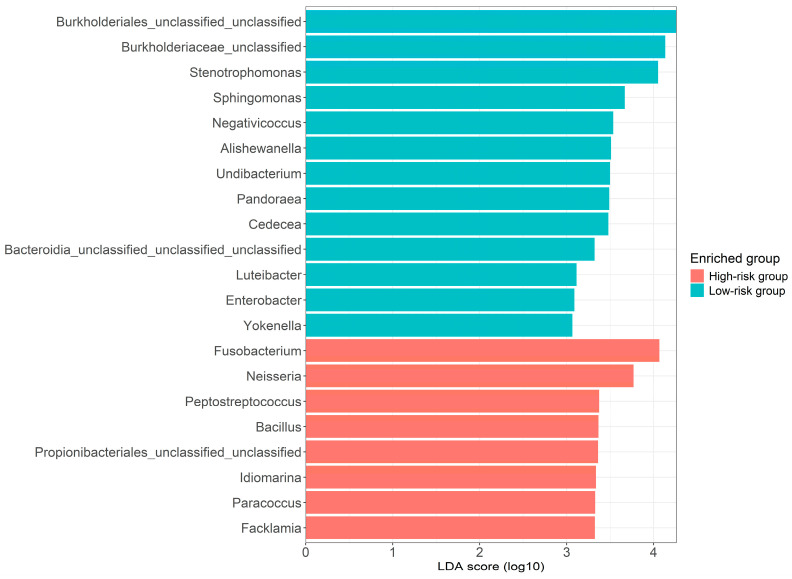
Linear discriminant analysis Effect Size (LEfSe) analysis indicated enriched genera in the high/low PSA risk groups. Green indicates taxa enriched in the Low-risk PSA group (*n* = 45), and orange indicates taxa enriched in the High-risk PSA group (*n* = 44).

**Figure 4 cancers-17-00070-f004:**
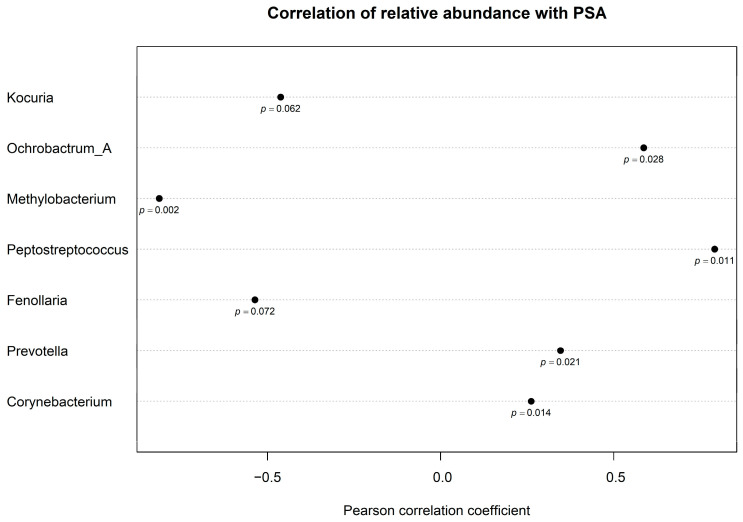
Dot chart with Pearson correlation coefficients between relative abundance of genera and PSA. All displayed genera had a correlation with PSA with a *p*-value < 0.1. The *p*-values are displayed below each point.

**Table 1 cancers-17-00070-t001:** Summary of the anaerobic bacterial species detected in urine samples from the culturomics cohort (*n* = 39). MA: microaerophilic bacteria, OA: obligate anaerobic bacteria, PSA: prostate-specific antigen level.

Samples	MA Status	OA Status	PSA (ng/mL)	
MA/OA-Negative	Neg	Neg	Mean = 4.75	
*n* = 22 samples			Median = 3.40	
MA-positive sample ID	MA status	OA status	PSA (ng/mL)	Bacterial species
8JL	Pos	Neg	12	*Brevibacterium casei*
17SR	Pos	Neg	5.36	*Actinomyces neuii*
3 PO	Pos	Neg	7.31	*Actinomyces neuii*
13 PL *	Pos	Pos	7.4	*A. turicecis*, *B. paucvorans* + 2 OA
9DG	Pos	Neg	4.54	*Cutibacterium acnes*
13EA	Pos	Neg	2.43	*Cutibacterium acnes*
*n* = 6 samples			Mean = 6.51 Median = 6.34	
OA-positive sample ID	MA status	OA status	PSA (ng/mL)	Bacterial species
2 PH	Neg	Pos	4.54	*Peptoniphilus harei*, *Veillonella* montpellierensis
12 ST	Neg	Pos	26	*Peptoniphilus harei*
16 WP	Neg	Pos	26	*Peptoniphilus harei*, *Peptostreptococcus* anaerobius, *Finegoldia* magna
13 PL *	Neg	Pos	7.4	*Veillonella parvula*, *Actinobaculum schaalii* + 2 MA
4 HS	Neg	Pos	6.1	*Peptoniphilus harei*, *Fusobacterium nucleatum*, *Fusobacterium gondiaformans*, *Actinobaculum schaalii*
3 PSS	Neg	Pos	1.5	*Peptostreptococcus anaerobius*
4 NC	Neg	Pos	9.64	*Peptoniphilus harei*
11 ND	Neg	Pos	13.6	*Veillonella dispar*
15 AS	Neg	Pos	1.12	*Veillonella* ratti, *Prevotella buccalis*
18 HS	Neg	Pos	2.54	*Peptoniphilus harei*, *Peptoniphilus lacrimalis*
19 PS	Neg	Pos	7.75	*Prevotella melaninogenica*
21 JF	Neg	Pos	4.78	*Peptoniphilus harei*
*n* = 12 samples			Mean = 9.25 Median = 6.75	

* Patient sample with both OA and MA.

**Table 2 cancers-17-00070-t002:** Cuzick test (extension for the Wilcoxon rank-sum test) for trend in 3 ordered groups (corrected for ties) with simulated *p*-value and confidence intervals based on 2000 replicates. “Neither”: MA/OA-negative samples, “Microaerophilic”: MA-positive samples, “Obligate Anaerobes”: OA-positive samples, CI: confidence interval for the simulated *p*-value.

Bacteria Present	Rank Order	N	Median PSA (ng/mL)
Neither	1	22	3.54
Microaerophilic	2	6 *	6.34
Obligate Anaerobes	3	12	6.75
Z			2.00
*p*-value			0.034
CI			0.026–0.042

* The sum of the samples is 40, as one of the samples is used in both the MA and OA group.

**Table 3 cancers-17-00070-t003:** Sample frequency and mean PSA for OA-positive and OA-negative groups. Culturomics samples were stratified based on the presence of one or more OAs. The six OA genera used to stratify PROVENT samples are *Finegoldia*, *Fusobacterium*, *Prevotella*, *Peptoniphilus_A*, *Peptostreptococcus*, and *Veillonella_A*. OA threshold: pooled relative abundance threshold for a PROVENT sample to be considered as OA-positive. The displayed *p*-value is from Welch’s two-sample *t*-test.

	OA-Positive	OA-Negative	OA Threshold
Culturomics cohort (*n* = 39)	30.8% (*n* = 12)	69.2% (*n* = 27)	-
PSA (mean ng/mL)	9.25 (*p* = 0.132)	5.04	-
PROVENT series (*n* = 89)	23.6% (*n* = 21)	76.4% (*n* = 68)	≥10% of all genera
PSA (mean ng/mL)	8.48 (*p* = 0.133)	7.15	≥10% of all genera
PROVENT series (*n* = 89)	21.3% (*n* = 19)	78.7% (*n* = 70)	≥15% of all genera
PSA (mean ng/mL)	8.75 (*p* = 0.082)	7.11	≥15% of all genera
PROVENT series (*n* = 89)	15.7% (*n* = 14)	84.3 (*n* = 75)	≥20% of all genera
PSA (mean μg/L)	9.47 (*p* = 0.006)	7.08	≥20% of all genera

**Table 4 cancers-17-00070-t004:** Crosstabulation of PROVENT samples based on OA status and high/low-risk * PSA groups. Bacteria: *Finegoldia*, *Fusobacterium*, *Prevotella*, *Peptoniphilus_A*, *Peptostreptococcus*,and *Veillonella_A*.

**Number of samples (pooled relative abundance threshold: ≥10%)**	**Obligate anaerobe** **Negative ***	**Obligate anaerobe** **Positive ***
Low-risk PSA group	40	5
High-risk PSA group	28	16
Fisher’s exact test		
*p*-value	0.006	
Odds ratio	4.493	
95 CI	1.369–17.57	
**Number of samples (pooled relative abundance threshold: ≥15%)**	**Obligate anaerobe** **negative**	**Obligate anaerobe** **positive**
Low-risk PSA group	42	3
High-risk PSA group	28	16
Fisher’s exact test		
*p*-value	<0.001	
Odds ratio	7.82	
95 CI	1.980–45.71	
**Number of samples** **(pooled relative abundance threshold: ≥20%)**	**Obligate anaerobe** **negative**	**Obligate anaerobe** **positive**
Low-risk PSA group	44	1
High-risk PSA group	31	13
Fisher’s exact test		
* p * -value	<0.001	
Odds ratio	17.97	
95 CI	2.462–798.2	

* Stratification of samples to high/low-risk PSA groups was done based on median PSA of the PROVENT samples: PSA > 7.1 ng/mL—High-risk PSA group, PSA ≤ 7.1 ng/mL—Low-risk PSA group.

**Table 5 cancers-17-00070-t005:** Selected obligate anaerobe genera and mean PSA levels in PROVENT samples (*n* = 89). Stratification of samples to OA-positive/negative groups was done based on the relative abundance (RA ≥10%, ≥5%, or >0%) of the corresponding genus. The displayed *p*-value is from Welch’s two-sample *t*-test. *p* = n/a: sample size is 0 or it is too small to perform a *t*-test.

	OA-Positive (≥10% RA)	OA-Negative	OA-Positive (≥5% RA)	OA-Negative	OA-Positive (>0% RA)	OA -Negative
*Peptostreptococcus*	0% (*n* = 0)	100% (*n* = 89)	0% (*n* = 0)	100% (*n* = 89)	10.11% (*n* = 9)	89.89% (*n* = 80)
PSA (mean ng/mL)	n/a	7.46	n/a	7.46	9.73 (*p* = 0.033)	7.21
*Fusobacterium*	2.25% (*n* = 2)	97.75% (*n* = 87)	2.25% (*n* = 2)	97.75% (*n* = 87)	11.24% (*n* = 10)	88.76% (*n* = 79)
PSA (mean ng/mL)	11 (*p* = 0.253)	7.38	11 (*p* = 0.253)	7.38	10.04 (*p* = 0.009)	7.14
*Prevotella*	19.1% (*n* = 17)	80.9% (*n* = 72)	24.72% (*n* = 22)	75.28% (*n* = 67)	49.44% (*n* = 44)	50.56% (*n* = 45)
PSA (mean ng/mL)	8.55 (*p* = 0.197)	7.21	8.02 (*p* = 0.409)	7.28	7.07 (*p* = 0.265)	7.84
*Peptoniphilus_A*	1.12% (*n* = 1)	98.88% (*n* = 88)	2.25% (*n* = 2)	97.75% (*n* = 87)	48.31% (*n* = 43)	51.69% (*n* = 46)
PSA (mean ng/mL)	8.2 (*p* = n/a)	7.46	8.30 (*p* = 0.022)	7.44	6.94 (*p* = 0.144)	7.95
*Veillonella_A*	0% (*n* = 0)	100% (*n* = 89)	0% (*n* = 0)	100% (*n* = 89)	2.25% (*n* = 2)	97.75% (*n* = 87)
PSA (mean ng/mL)	n/a	7.46	n/a	7.46	8.10 (*p* = 0.513)	7.45
*Finegoldia*	0% (*n* = 0)	100% (*n* = 89)	2.25% (*n* = 2)	97.75% (*n* = 87)	51.69% (*n* = 46)	48.31% (*n* = 43)
PSA (mean ng/mL)	n/a	7.46	7.35 (*p* = 0.747)	7.47	7.23 (*p* = 0.490)	7.71

**Table 6 cancers-17-00070-t006:** Effect of the >0% relative abundance of 6 genera in the culturomics series and the 2 significant hits from the correlation analysis (*Methylobacterium* and *Ochrobactrum_A*) on PSA (multiple linear regression analysis).

Predictor	Coefficient	Standard Error	t Value	*p*-Value
Intercept	6.974	0.396	17.604	<0.001 ***
*Peptostreptococcus*	1.715	0.621	2.76	0.007 **
*Ochrobactrum_A*	8.974	3.374	2.659	0.009 **
*Prevotella*	0.03	0.025	1.194	0.236
*Methylobacterium*	−2.457	1.381	−1.78	0.079 *
*Peptoniphilus_A*	0.013	0.045	0.291	0.772
*Fusobacterium*	0.04	0.071	0.572	0.569
*Finegoldia*	−0.144	0.325	−0.444	0.658
*Veillonella_A*	0.498	0.9	0.554	0.581
Model fit statistics				
Residual standard error: 3.005 on 80 degrees of freedom				
Multiple R-squared: 0.225				
Adjusted R-squared: 0.147				
F-statistic: 2.894 on 8 and 80 degrees of freedom,				
*p*-value: 0.007				

* *p*-value < 0.1, ** *p*-value < 0.01, *** *p*-value < 0.001.

## Data Availability

The 16s rDNA sequencing data for this study have been deposited in the Genome Sequence Archive (https://ngdc.cncb.ac.cn/gsa/, accessed on 26 November 2024) under the BioProject PRJCA032503. Annotated read counts per ASV per sample are available in Appendix A.

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
