# Peer review of "Exploring the Link Between Obligate Anaerobe-Related Dysbiosis and Prostate Cancer Development: A Pilot Study"

_cancers, 2024, doi:10.3390/cancers17010070_

Round 1
Reviewer 1 Report
Comments and Suggestions for Authors
The manuscript entitled, ‘Exploring the Link Between Obligate Anaerobes-Related Dysbiosis and Prostate Cancer Development: A Pilot Study’ intends to explore the link between obligate anaerobes-related dysbiosis and prostate cancer development. For that, this study has analyzed post-rectal urine samples from 89 men with prostate cancer via 16s rDNA sequencing. The manuscript is well structured, the analysis is sound, and it presents an insightful contribution to the field. However, minor revisions are suggested to enhance clarity and ensure precision in certain areas. Once the minor revisions are addressed, I believe the manuscript will be ready for publication in the Journal of Cancers.
1. The #Introduction section lacks the storyline and may not be suitable for general readers. For instance, the first paragraph (#Line_433-443) of #Conclusion section could be included in the Introduction section.
2. The name of the microbes should be written in italics throughout the manuscript.
3. The quality of the figures must be improved for better clarity and readability.
4. The legends of the figures (presented in supplementary materials) should be comprehensive and clearly explained (e.g., the heatmaps).
5. The limitations and important findings of the study should be distinctly mentioned in the #Conclusion section.
Author Response
Dear Reviewer,
Thank you very much for your time and effort in reviewing our manuscript “Exploring the Link Between Obligate Anaerobes-Related Dysbiosis and Prostate Cancer Development: A Pilot Study” that is currently under consideration for publication in Cancers. Please find below our responses to your comments:
Comment 1. The #Introduction section lacks the storyline and may not be suitable for general readers. For instance, the first paragraph (#Line_433-443) of #Conclusion section could be included in theIntroduction section.
Response 1: The Introduction section has been enriched with a clear explanation about the story and issues behind prostate cancer (lines 47-68). The first paragraph of the Conclusion section is now embedded in the Introduction as per your suggestion (lines 71-81)
Comment 2.The name of the microbes should be written in italics throughout the manuscript.
Response 2: We have now converted all microbe names to italics.
Comment 3.The quality of the figures must be improved for better clarityand readability.
Response 3: We have now converted all figures to a higher quality (600 dpi).
Comment 4.The legends of the figures (presented in supplementary materials) should be comprehensive and clearly explained (e.g.,the heatmaps).
Response 4: We have amended the legends of the figures adding a clearer explanation. Figure S3 (heatmap) was also explained more properly.
Comment 5. The limitations and important findings of the study should be distinctly mentioned in the #Conclusion section.
Response 5: We have added a few paragraphs in the Discussion section describing the important finding and limitations of our study (lines 490 – 517)
We look forward to hearing from you in due time regarding our submission and to respond to any further questions and comments you may have.
Yours sincerely
Dr. Belinda Nedjai

Reviewer 2 Report
Comments and Suggestions for Authors
This study investigates the potential link between specific bacteria and prostate cancer (PC), focusing on how bacterial dysbiosis might contribute to elevated prostate-specific antigen (PSA) levels, a biomarker often used in prostate cancer screening. This manuscript provides a comprehensive explanation and a good remark in this avenue. Some improvements and suggestions have been provided to enhance its quality:
-If it is not necessary for the journal format, it is pointless to give a simple summary where an abstract is already written.
Please give references at the end of the sentence; for instance, you should fix the placement of 8-13, 15, 17, 18, and 21 references in the introduction part.
- I suggest authors expand the introduction part of your manuscript. Detailed investigations are present in your study, and you should mention related aspects to enrich the explanation.
- A single abbreviation should be used for prostate cancer in whole aricle.
- Please write the species names in italics throughout the whole text.
- The use of any website bridges is not recommended; instead, references should be employed where appropriate.
- Prostate cancer should be explained with more details in the introduction section.
- In lines 54 and 169, the reference should be at the end of the sentence.
-Line 167, please remove the underline of the words, which are “high-risk group and low-risk group.”.
-What is MA bacteria? Please explain this abbreviation.
- The MALDI-TOF abbreviation should be explained.
-In line 212, the sentence should be written correctly with words in line 213.
-Although the discussion and conclusion parts are well explained and supported by references, what will this study contribute to future studies, or what is the evaluation of this issue from a future perspective? A section answering these questions should be added.
Author Response
Dear Reviewer,
Thank you very much for your time and effort in reviewing our manuscript “Exploring the Link Between Obligate Anaerobes-Related Dysbiosis and Prostate Cancer Development: A Pilot Study” that is currently under consideration for publication in Cancers. Please find below our responses to your comments:
Comment 1. If it is not necessary for the journal format, it is pointless to give a simple summary where an abstract is already written.
Response 1: We understand that two summaries might seem redundant, however it is necessary for the journal format to include a simple summary before the abstract according to the author guidelines (https://www.mdpi.com/journal/cancers/instructions). According to the guidelines “Submissions without a simple summary will be returned directly.”
Comment 2. Please give references at the end of the sentence; for instance,you should fix the placement of 8-13, 15, 17, 18, and 21 referencesin the introduction part.
Response 2: We have revised the manuscript and put references at the end of the sentences. All references in the Introduction section are now at the end of the corresponding sentences.
Comment 3. I suggest authors expand the introduction part of your manuscript. Detailed investigations are present in your study, and you should mention related aspects to enrich the explanation.
Response 3: Thank you for your suggestion. We have expanded the Introduction part of our manuscript mentioning related aspects of prostate cancer (lines 48-81).
Comment 4. A single abbreviation should be used for prostate cancer in whole aricle.
Response 4: Thank you for noticing that. We have now changed all abbreviations of prostate cancer to “PCa”
Comment 5. Please write the species names in italics throughout the whole text.
Response 5: We have now converted all microbe names to italics.
Comment 6. The use of any website bridges is not recommended; instead,references should be employed where appropriate.
Response 6: We have now removed the website for Clinical Microbiomics (line 141) and the PROVENT clinical trial (line 133). We replaced it with the ISRCTN ID instead.
Comment 7. Prostate cancer should be explained with more details in the introduction section.
Response 7: The revised Introduction has now a detailed explanation about prostate cancer (lines 48-58)
Comment 8. In lines 54 and 169, the reference should be at the end of the sentence.
Response 8: These references have now been placed at the end of the sentence. (new lines: 84 and 206)
Comment 9. Line 167, please remove the underline of the words, which are“high-risk group and low-risk group.”.
Response 9: The underline of the words has been removed (new line: 202)
Comment 10. What is MA bacteria? Please explain this abbreviation.
Response 10: We abbreviate microaerophilic bacteria as “MA bacteria”. This abbreviation is now mentioned in the Introduction section (line 70).
Comment 11. The MALDI-TOF abbreviation should be explained.
Response 11: The MALDI-TOF abbreviation is now explained in the Introduction section (line 95)
Comment 12. In line 212, the sentence should be written correctly with words in line 213.
Response 12: Section 3.1 “Culturomics cohort” (new lines 245-246) has been revised and correctly written. Part of the previous sentence is now added to the description of Table 1.
Comment 13. Although the discussion and conclusion parts are well explained and supported by references, what will this study contribute to future studies, or what is the evaluation of this issue from a future perspective? A section answering these questions should beadded.
Response 13: A few paragraphs have been added in the Discussion section explaining the strength of this study and how its results will contribute to future studies and to future clinical decisions concerning prostate cancer (lines 490-516)
We look forward to hearing from you in due time regarding our submission and to respond to any further questions and comments you may have.
Yours sincerely
Dr. Belinda Nedjai

Reviewer 3 Report
Comments and Suggestions for Authors
Exploring the Link Between Obligate Anaerobes-Related 2 Dysbiosis and Prostate Cancer Development: A Pilot Study, looks at 2 classes of bacteria, microaerobic and obligate anaerobes to understand the relationship between PSA levels and cancer. This is an attempt to fill in gaps from earlier work by this group using culture methods and long-stand claims about the link between certain bacteria and prostate cancer, although not confirming the original claim to any degree, while nominating others. This gets a little confusing since both are addressed and the PSA side shoves the immediate cancer discussion aside. Perhaps this could be better prefaced in the introduction. However, the point is made that the bacteria driven inflammation might be a precursor to cancer. The PSA results in response to certain classes of bacteria might be an early warning of predisposition to cancer. Because of this, the authors suggest that in light of natural resistance of the anaerobes to antibiotics and increasing antibiotic resistance, immune checkpoint therapies might be used.
The results are presented reasonable straight forward, although I find little value in Fig. 3, with only one genus from tables 1 and 2 showing up. Like wise, fig. 4 and table 6. could be trimmed.
Author Response
Dear Reviewer,
Thank you very much for your time and effort in reviewing our manuscript “Exploring the Link Between Obligate Anaerobes-Related Dysbiosis and Prostate Cancer Development: A Pilot Study” that is currently under consideration for publication in Cancers. Please find below our responses to your comments:
Comment 1. The results are presented reasonable straight forward, although I find little value in Fig. 3, with only one genus from tables 1 and 2 showing up. Like wise, fig. 4 and table 6. could be trimmed.
Response 1: Thank you for pointing this out. We appreciate how the figures that do not include all genera from tables 1 and 2 might seem unnecessary. However Figure 3 and Figure 4 present the results of a strict data-driven analysis (LEfSe and Pearson correlation respectively) that included all species detected in the PROVENT cohort. Moreover Table 6 presents the results of the multiple regression analysis and we believe that including the genera from the culturomics cohort together with the top hits of the data-driven analysis is the best way to provide perspective about their effect on PSA. For that reason we have now added a paragraph in the Discussion section explaining their importance (lines 495-501)
We look forward to hearing from you in due time regarding our submission and to respond to any further questions and comments you may have.
Yours sincerely
Dr. Belinda Nedjai

Round 2
Reviewer 2 Report
Comments and Suggestions for Authors
The authors completed the revision appropriately.
Reviewer 3 Report
Comments and Suggestions for Authors
The additions to the introduction and discussion improve the paper to publishable levels.